# MaSS: Multi-attribute Selective Suppression

## Abstract

The recent rapid advances in machine learning technologies largely depend on the vast richness of data available today. Along with the new services and applications enabled by those machine learning models, various governmental policies are put in place to regulate such data usage and protect people's privacy and rights. As a result, data owners often opt for simple data obfuscation (e.g., blur people's faces in images) or withholding data altogether, which leads to severe data quality degradation and greatly limits the data's potential utility. Aiming for a sophisticated mechanism which gives data owners fine-grained control while retaining the maximal degree of data utility, We propose *Multi-attribute Selective Suppression*, or *MaSS*, a general framework for performing precisely targeted data surgery to simultaneously *suppress* any selected set of attributes while *preserving* the rest for downstream machine learning tasks. MaSS learns a data modifier through adversarial games between two sets of networks, where one is aimed at suppressing selected attributes, and the other ensures the retention of the rest of the attributes. We carried out an extensive evaluation of our proposed method using multiple datasets from different domains including facial images, voice audio, and video clips, and obtained highly promising results in MaSS' generalizability and capability of drastically suppressing targeted attributes while imposing virtually no impact on the data's usability in other downstream machine learning tasks.

## 1 Introduction

The recent rapid advances in machine learning (ML) can be largely attributed to powerful computing infrastructures as well as the availability of large-scale datasets, such as ImageNet1K (Deng et al., 2009) for computer vision, WMT (Foundation, 2019) for neural machine translation, and LibriLight (Kahn et al., 2020) for speech recognition. Studies have shown that ML models trained on large-scale datasets can usually prove effective in many additional downstream tasks (Brown et al., 2020). On the other hand, ethical concerns have been raised surrounding proper data usage in issues like data privacy (Liu et al., 2021), data minimization (Goldsteen et al., 2021), etc. Therefore, if there are more data available and can be used without worrying whether or not the data is handled properly, the ML models can be further improved by more data and help the ML community to advance on many domains.

Attempting to balance between model performance and proper data usages, a common approach usually taken is to simply modify the data to remove its "sensitive" attributes, and experimentally demonstrate that the targeted sensitive attributes are indeed removed. What's crucially important but usually omitted here, however, is the preservation of the "total utility" of the data, because the suppression operation oftentimes also negatively impact, or even completely destroy, the other "non-sensitive" attributes, hence greatly damaging the dataset's potential future utility. For example, DeepPrivacy (Hukkelås et al., 2019) *is* able to demonstrate its privacy protection capability, but the modified data it produces can no longer be utilized for additional downstream tasks like sentiment analysis, age detection, or gender classification. Since data is one of the main driving forces for the rapid advancement of machine learning research, we argue that the ideal scenario would be to have the flexibility of selecting an arbitrary set of attributes and only suppressing *them* while leaving all the other attributes completely intact. In this way, the community could unleash the potential utility of the modified data to develop more advanced algorithms.

Towards this exact goal, we present Multi-attribute Selective Suppression (or MaSS) in this paper, to enable such capability of precise attributes suppression for multi-attribute datasets. The high-level

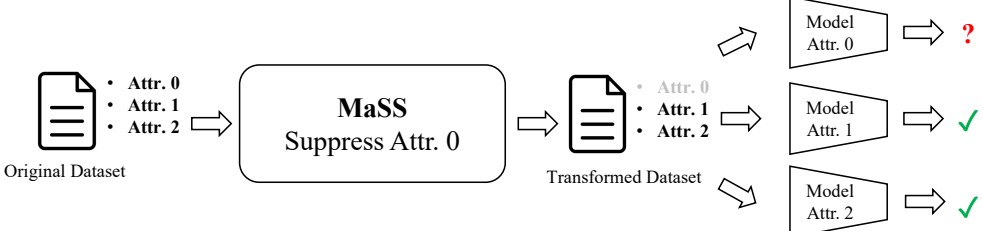

Figure 1: MaSS is able to precisely target any selected attributes in a multi-attribute dataset for suppression while leaving the rest of the attributes intact for any potential downstream ML-based analytic tasks. For example as illustrated by the diagram, when operating on the original multi-attribute dataset and configured to suppress Attr. 0, MaSS is able to transform the dataset such that the model for detecting Attr. 0 is unable to reliably detect Attr. 0 from the transformed data, while the models for Attr. 1 and 2 are not affected.

objective of MaSS is also illustrated in Figure 1, where MaSS is configured to suppress Attr. 0 without knowing in advance that Attr. 1 and 2 will be used for downstream tasks. After the data transformation performed by MaSS, Attr. 0 becomes suppressed, nondetectable by its corresponding machine learning model, but at the same time, Attr. 1 and 2 are left intact, and still can be extracted by their corresponding ML models. As a concrete example, suppose we are working with a facial-image dataset which contains attributes like age, gender, and sentiment, where, let us assume, age and gender are considered sensitive. Then, MaSS would transform this facial-image dataset such that age and gender information could no longer be not be inferred by the corresponding ML models, whereas sentiment information could still be extracted from the transformed data.

The contributions of our work are threefolds,

1. We propose the novel MaSS framework to enable the powerful flexibility of precise suppression of arbitrary, selective data attributes.

2. We employ multiple learning mechanisms in MaSS to enable its attribute-specific as well as generic feature preservation capabilities, which help it achieve satisfactory data utility protection both with and without the prior knowledge about downstream tasks.

3. We thoroughly validate MaSS using a wide range of multi-attribute datasets, including image, audio, and video. All our results demonstrate MaSS' strong performance in its intended selective attribute suppression and preservation.

## 2 RELATED WORKS

**Data Privacy.** A large body of work has studied methods applying generative adversarial networks (GANs) to generate and modify facial features in images, so these identity-related sensitive features can be de-identified. DeepPrivacy (Hukkelås et al., 2019) proposed to use a conditional generative adversarial network to generate realistic anonymized faces, while considering the existing background and a sparse pose annotation. To further ensure the face anonymization using GAN-based methods, CIAGAN (Maximov et al., 2020) proposed an identity control discriminator to control which fake identity is used in the anonymizaiton process by introducing an identity control vector. Instead of generating the entire faces for anonymization, Li et al. proposed to apply conditional GAN to only identify and modify the five identity-sensitive attributes. To also enable face anonymization with the selected semantic attributes manipulation, PI-Net (Chen et al., 2021) proposed to generate realistic looking faces with the selected attributes preserved. The above works usually focuses on suppression only while the future data utilities are not considered. Our approach not only suppresses the attributes but also preserves the data utility concurrently. On the other hand, SPAct tried to suppress the multiple attributes in a video through contrastive learning while preserving the utility for action recognition; however, their approach lacks the flexibility to handle individual attributes but can only process all attributes at once and limits to the action recognition dataset; while our method is fully configurable and validated in different data domains. Moriarty et al. proposed the method to suppress the biometric information while preserving its utility; however, their approach requires the information of downstream task while our method does not.

**Multi-Attribute Learning.** Data could contain multiple attributes. For recognition, using one model for each attribute might provide good recognition accuracy but might not scale. Multi-attribute learning aims for resolving this problem by learning underlying relationship among attributes such that the model can adaptively handle different attributes (Huang et al., 2018; Cheng et al., 2018). In this paper, we validated our method with the single-attribute models. The multi-attribute model can also be used in our method by replacing multiple single-attribute models.

**Dataset Distillation/Condensation.** Dataset distillation and condensation aim to create a smaller version of a large dataset which can be used to train a model whose performance can be as good as training on the original large dataset. Therefore, training can be relatively quicker, e.g., Neural Architecture Search (NAS) methods require a lot of iterations of a whole dataset to find out the best model. Otherwise, NAS usually needs to use a proxy dataset/model for the approximation results from large dataset/model (Zhao et al., 2021; Cazenavette et al., 2022; Wang et al., 2022). On the other hand, a recent work shows the condensed dataset also conceals some attributes from the original data (Dong et al., 2022) but still remains effective for the original task. In contrast to these approaches, our proposed method tries to keep the truthfulness of data as much as possible, so we do not reduce the amount of data. Moreover, our method is designed to preserve the generic features rather than the specific task, which could be covered by the generic feature we preserved.

**Self-supervised/Contrastive Learning.** Self-supervised learning has been explored by the community to allow a machine learning model to learn a good data representation by designing pretext tasks instead of human annotations (Feng et al., 2019), or contrastive learning which tries to maximize the agreement between positive pairs (Chen et al., 2020; He et al., 2020; Grill et al., 2020), or clustering-based methods to generate pseudo labels for data (Caron et al., 2020; 2018), or mask autoencoder to predict the masked patches by the remaining patches (He et al., 2022). Then, they show that the feature representations are usually good for many different downstream tasks. Our goal is to keep as much information when suppressing the selected attributes without having prior knowledge; thus, self-supervised and contrastive learning methods facilitate our requirement to extract generic features without label information from downstream tasks.

## 3 PROPOSED METHOD

### 3.1 PROBLEM DEFINITION

Consider a multi-attribute dataset $\mathbf{X}$ with size $N$, and the set $A$ of different attributes, where each data point $\mathbf{x}$'s value for attribute $a \in A$ is $a_{\mathbf{x}}$. Each of the different attributes can be learned by inferencing on $\mathbf{X}$, hence our objective is to transform $\mathbf{X}$ to suppress any selective subset of attributes $S \subseteq A$ such that no attributes in $S$ can be reliably inferred from the transformed dataset $\mathbf{X}'$. At the same time, we need to make sure the rest of the attributes $R = A \setminus S$ are preserved and can still be inferred from $\mathbf{X}'$.

In practice, when a data owner would like to transform their data, we assume the subset $S$ of attributes targeted for suppression is always predetermined. However, it is not always known in advance what the entire set $A$ of attributes are, and consequently which set $R$ of attributes need to be preserved. Therefore, for generalizability, we consider the case of an unknown $R$ at the time of data transformation. Of course, if $R$ happens to be given, our solution needs to be able to take advantage of this extra information as well.

### 3.2 PROPOSED FRAMEWORK

We propose a Generative Adversarial Network (GAN)-based solution in tackling the data transformation problem. Our framework consists of three major components: a *Data Modifier*, a *Suppression Branch*, and a *Preservation Branch*, as depicted in Fig. 2. The data modifier $G$ is the generator while both branches are the discriminators in the GAN framework. The data modifier tries to generate new data such that the similarity between original data and modified data are maximized and minimized via the suppression branch and the preservation branch, respectively. In a nutshell, the data modifier learns a transformation that is to be applied to the original data vectors, where the learned transformation is jointly regularized by both the suppression and preservation branches to

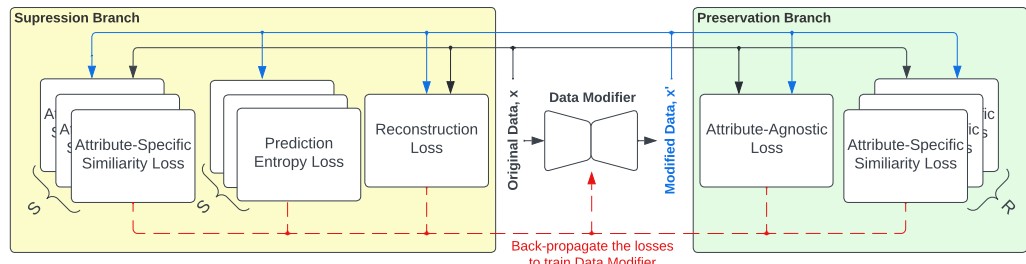

Figure 2: **Overview of MaSS.** MaSS contains three components: the data modifier, the suppression branch, and the preservation branch. The data modifier $G$ is trained by optimizing the losses in both the suppression and the preservation branches.

ensure all targeted attributes in $S$ are indeed suppressed in the transformed data, while all other attributes, explicitly specified or not, are left intact as much as possible. We next discuss all three components in more detail.

### 3.2.1 DATA MODIFIER

We design the data modifier to be conditional on its original input in learning instance-specific transformations via a multi-layer perceptron (MLP) with a residual shortcut. In other words, based on its input $\mathbf{x}$, the data modifier $G$ learns an additive modification to be applied to $\mathbf{x}$. Therefore, the transformed data stays in the same embedding space as the original, which leads to faster optimization convergence. Specifically, for a normalized original data vector $\mathbf{x}$, its normalized transformed version is computed as

$$\mathbf{x}' = G(\mathbf{x}) := n(\mathbf{x} + n(\text{MLP}(\mathbf{x}))), \tag{1}$$

where $n(\cdot)$ is the normalization function $n(\mathbf{v}) = \begin{cases} \frac{\mathbf{v}}{||\mathbf{v}||}, & ||\mathbf{v}|| \neq 0 \\ 0, & \text{o.w.} \end{cases}$.

During the optimization, the parameters in the data modifier are trained by back-propagating the losses designed in both branches.

### 3.2.2 SUPPRESSION BRANCH

The suppression branch is designed to make the targeted attributes in $S$ as unrecognizable as possible. It utilizes the corresponding set of inference models pretrained on the original dataset $\mathbf{X}$. Each pretrained model corresponds to a specific attribute $s \in S$ and is composed of a feature extractor $F^s$, which converts raw input data to a feature vector $\mathbf{z}_\mathbf{x}^s = F^s(\mathbf{x})$, and a projector $P^s$, which maps the features to the attribute's label $\mathbf{p}_\mathbf{x}^s = P^s(\mathbf{z}_\mathbf{x}^s)$, where $\mathbf{z}_\mathbf{x}^s$ is the feature representation and $\mathbf{p}_\mathbf{x}^s$ is the prediction logit.

During the training of the data modifier, the suppression branch guides the data modifier to degrade these pretrained models' recognition accuracies on the transformed data $\mathbf{X}'$. It does so by either measuring the similarity of features from the pretrained models or by comparing the prediction results against the ground truth labels, and then adding a corresponding penalty. Specifically, for each targeted attribute $s \in S$, the feature-similarity loss function is defined as

$$L_{\text{sim}}^s = w^s \cdot \frac{1}{N} \sum_{\mathbf{x} \in \mathbf{X}} \text{sim}^s(\mathbf{x}', \mathbf{x}), \tag{2}$$

where $w^s$ is the weight for attribute $s$ and $\text{sim}^s(\cdot, \cdot)$ defines a similarity measure. For example, we can use the cosine similarity between feature vectors $\text{sim}_{\cos}^s(\mathbf{x}', \mathbf{x}) = \cos(\mathbf{z}_{\mathbf{x}'}^s, \mathbf{z}_\mathbf{x}^s)$, or the negative KL-divergence between the original and the transformed logits $\text{sim}_{\text{KL}}^s(\mathbf{x}', \mathbf{x}) = -D_{\text{KL}}(\mathbf{p}_{\mathbf{x}'}^s \| \mathbf{p}_\mathbf{x}^s)$, or the cross-entropy loss between the transformed logits and the attribute's ground truth value $\text{sim}_{\text{CE}}^s(\mathbf{x}', \mathbf{x}) = -D_{\text{KL}}(\mathbf{p}_{\mathbf{x}'}^s \| \mathbf{1}_{s_\mathbf{x}})$, where $s_\mathbf{x}$ is $\mathbf{x}$'s ground truth value for attribute $s$ and $\mathbf{1}_{s_\mathbf{x}}$ is the corresponding one-hot vector.

The above similarity-based metrics only help guide the data modifier to lead the pretrained models towards incorrect predictions about $S$ on the transformed data. However, in terms of attribute suppression, an even stronger condition would be to reduce such predictions to random guesses. Therefore, we introduce an additional loss term to maximize the entropy $H$ of the predictions about $S$ on the transformed data such that the predictions look like random guesses. Combined with Eq. 2, the total loss on the suppression branch is

$$L^s = w^s \cdot \frac{1}{N} \sum_{\mathbf{x} \in \mathbf{X}} \text{sim}^s (\mathbf{x}', \mathbf{x}) - h^s \cdot \frac{1}{N} \sum_{\mathbf{x} \in \mathbf{X}} H \left( \mathbf{p}_{\mathbf{x}'}^s \right), \tag{3}$$

where $h^s$ is the weight for the entropy loss.

Moreover, to regularize the learned data transformation to be as small as possible, we also add an $L2$ reconstruction loss term in the suppression branch

$$L_{\text{rec}} = w_{\text{rec}} \cdot \frac{1}{N} \sum_{\mathbf{x} \in \mathbf{X}} ||\mathbf{x}' - \mathbf{x}||_2. \tag{4}$$

### 3.2.3 PRESERVATION BRANCH

While the suppression branch guides the data modifier to decrease the confidence of machine learning models on certain attributes $S$ from the data, the preservation branch is responsible for guarding the data against said suppression and erasure such that maximum utility can be preserved through the transformation, in the sense that all the attributes $R = A \setminus S$ not targeted by the suppression branch should remain recognizable from the transformed data $\mathbf{X}'$, just as they were from the original data $\mathbf{X}$. In the proposed method, we design two types of losses: one is *attribute-agnostic* in the preservation branch, which is applicable to any attribute as the loss is defined in an agnostic way; another one is *attribute-specific*, which if the downstream tasks have been defined and we know the set of $R$, then we embedded them into the loss.

**Attribute-Agnostic.** As previously discussed, it is oftentimes difficult to identify in advance the set $A$ of *all* attributes associated with a dataset, which means the set $R$ might not necessarily be defined even when $S$ is specified. The absence of $R$ indicates that we have no knowledge about what attributes downstream tasks might want to detect from the transformed dataset.

To tackle this *attribute-agnostic* scenario, we employ self-supervised techniques, where a generic feature representation of the data is learned without any specific attribute or task information. For our preservation branch in particular, we adopt the SimCLR (Chen et al., 2020) contrastive-learning-based approach to pretrain an extractor $F^{r_*}$ and a projector $P^{r_*}$ that maximize the similarity between embedding pairs originated from the same data point (i.e., positive pairs) and minimize that of different data points (i.e., negative pairs). Instead of considering two different transformations like SimCLR does, we treat an original data point $\mathbf{x}$ and its transformed version $\mathbf{x}'$ as a positive pair, and all the rest as negative. We enforce this relationship by applying normalized temperature-scaled cross entropy (NT-Xent) loss function introduced in SimCLR; therefore, semantically, the transformed data could preserve more generic features. For the original data point $\mathbf{x}$-based positive pair $(\mathbf{x}, \mathbf{x}')$, the NT-Xent loss $l_{\mathbf{x}}^{r_*}$ is

$$l_{\mathbf{x}}^{r_*} = - \log \frac{e^{\cos\left(\mathbf{p}_{\mathbf{x}}^{r_*}, \mathbf{p}_{\mathbf{x}'}^{r_*}\right)/\tau}}{\sum_{\mathbf{y} \in \mathbf{X}} \left[ e^{\cos\left(\mathbf{p}_{\mathbf{x}}^{r_*}, \mathbf{p}_{\mathbf{y}'}^{r_*}\right)/\tau} + \mathbb{1}_{\mathbf{y} \neq \mathbf{x}} \cdot e^{\cos\left(\mathbf{p}_{\mathbf{x}}^{r_*}, \mathbf{p}_{\mathbf{y}}^{r_*}\right)/\tau} \right]}, \tag{5}$$

and similarly for the transformed data point $\mathbf{x}'$-based positive pair $(\mathbf{x}', \mathbf{x})$, the NT-Xent loss $l_{\mathbf{x}'}^{r_*}$ is

$$l_{\mathbf{x}'}^{r_*} = - \log \frac{e^{\cos\left(\mathbf{p}_{\mathbf{x}'}^{r_*}, \mathbf{p}_{\mathbf{x}}^{r_*}\right)/\tau}}{\sum_{\mathbf{y} \in \mathbf{X}} \left[ e^{\cos\left(\mathbf{p}_{\mathbf{x}'}^{r_*}, \mathbf{p}_{\mathbf{y}}^{r_*}\right)/\tau} + \mathbb{1}_{\mathbf{y} \neq \mathbf{x}} \cdot e^{\cos\left(\mathbf{p}_{\mathbf{x}'}^{r_*}, \mathbf{p}_{\mathbf{y}'}^{r_*}\right)/\tau} \right]}, \tag{6}$$

where $\mathbf{p}_{\mathbf{x}}^{r_*} = P^{r_*}(F^{r_*}(\mathbf{x}))$ and $\mathbf{p}_{\mathbf{x}'}^{r_*} = P^{r_*}(F^{r_*}(\mathbf{x}'))$ are the attribute-agnostic logits computed from the original and transformed data points, respectively; $\tau$ controls the contribution proportion between the positive and the negative samples, with a higher $\tau$ value indicating a bigger contribution from

the positive samples, and vice versa; and $\mathbb{1}_{\mathbf{y} \neq \mathbf{x}}$ is the indicator function $\mathbb{1}_{\mathbf{y} \neq \mathbf{x}} = \begin{cases} 1, & \mathbf{y} \neq \mathbf{x} \\ 0, & \text{o.w.} \end{cases}$. The total loss for the preservation branch is thus aggregated across all data points

$$L^{r_*} = w^{r_*} \cdot \frac{1}{2N} \sum_{\mathbf{x} \in \mathbf{X}} (l_{\mathbf{x}}^{r_*} + l_{\mathbf{x}'}^{r_*}), \tag{7}$$

which enables the preservation branch to enforce the underlying representations of the transformed data points in $\mathbf{X}'$ to reach maximal agreement with their corresponding origins from $\mathbf{X}$ while maintaining their discriminative characteristics with each other.

**Attribute-Specific.** In addition to the attribute-agnostic case discussed above, there could also be *attribute-specific* situations, where the set $R$ of attributes that needs to be preserved is explicitly defined in advance for the data transformation. Therefore, we can pretrain the inference models on the original dataset $\mathbf{X}$ like in the suppression branch. To account for this extra information, $R$, we formulate the loss function similar to Eq. 2, except that here we want to maximize the similarities as opposed to minimizing them, as follows,

$$L^r = -w^r \cdot \frac{1}{N} \sum_{\mathbf{x} \in \mathbf{X}} \text{sim}^r (\mathbf{x}', \mathbf{x}), \tag{8}$$

where $w^r$ is the weight for each attribute $r \in R$.

Finally, collecting all the loss terms from the data modifier, the suppression branch, as well as the attribute-agnostic and attribute-specific components of the preservation branch, we have the overall optimization loss

$$L = \sum_{s \in S} L^s + L_{\text{rec}} + L^{r_*} + \sum_{r \in R} L^r. \tag{9}$$

## 4 EXPERIMENTAL EVALUATION

We next present and discuss our experimental evaluation of MaSS on its ability to perform selective attribute suppression and preservation when carrying out data transformations.

### 4.1 EXPERIMENTAL SETUP

We evaluated MaSS on three multi-attribute datasets of different domains, namely Adience (Eidinger et al., 2014) for facial images, AudioMNIST (Becker et al., 2018) for voice recordings, and PA-HMDB (Wu et al., 2020) for video clips. For all datasets, we converted all their raw data points to feature embeddings via state-of-the-art neural networks as the input $\mathbf{X}$. Table 1 lists the feature extractors used for each dataset, as well as the corresponding feature dimension and the architecture of MLP used in the data modifier. All our experiments were designed to examine MaSS' performance on attribute suppression and preservation quality, and not whether it could generate high quality synthetic data. Therefore, each transformed $\mathbf{x}'$ would stay in the same feature space as its corresponding $\mathbf{x}$. We next briefly introduce the datasets, as well as the training and evaluation protocols. More details can be found in Appx. A and B.

**Adience.** The Adience image dataset was originally published to help study the recognition of age and gender. Each image is also associated with a DataID. In total, the dataset used in our experiment contains 1,089 different DataIDs, 8 age groups, and 2 gender classes. We split the dataset into 2,815 for training and 2,525 for validation.

**AudioMNIST.** The AudioMNIST dataset contains audio recordings of spoken digits (0-9) in English from 60 speakers.

In addition to speakerID and spoken digits, the dataset also contain accent and gender attributes. Therefore, we use AudioMNIST as a 4-attribute dataset, namely speakerID, spoken digits, accent, and gender, with 60, 10, 18, and 2 classes, respectively. There are 30,000 audio clips in total. We split the data into 18,000, 6,000, and 6,000 for training, validation, and testing, respectively.

Table 1: Model configuration for each dataset.

| Dataset | Feature Extractor | Feature Dimension | MLP in Data Modifier |
|---|---|---|---|
| Adience | FaceNet (Schroff et al., 2015) | 512 | 512-256-128-256-512 |
| AudioMNIST | HuBERT-L (Hsu et al., 2021) | 1024 | 1024-256-128-256-1024 |
| PA-HMDB | R3D-18 (Tran et al., 2018) | 512 | 512-256-128-256-512 |

Table 2: Results on Adience under different configurations. The checkmark (✓) denotes that the particular loss is used in the optimization. DataID is selected as the suppression target.

| | Loss Configuration | | | | | Top-1 Accuracy (%) | | |
|---|---|---|---|---|---|---|---|---|
| | $L_{\text{rec}}$ | $L^{s_1}$ | $L^{r_*}$ | $L^{r_1}$ | $L^{r_2}$ | DataID ($s_1$) | Age ($r_1$) | Gender ($r_2$) |
| Original | - | - | - | - | - | 90.8 | 89.1 | 97.4 |
| MaSS | ✓ | ✓ | - | - | - | 0.0 | 33.8 | 73.5 |
| | ✓ | ✓ | ✓ | - | - | 0.6 | 78.5 | 95.7 |
| | ✓ | ✓ | ✓ | ✓ | - | 0.7 | 86.1 | 95.9 |
| | ✓ | ✓ | ✓ | ✓ | ✓ | 0.6 | 86.9 | 96.7 |

**PA-HMDB.** The PA-HMDB51 dataset is subset of HMDB51 and it contains 6 attributes: action, skin color, face, gender, nudity, and relationship (Kuehne et al., 2011; Wu et al., 2020). The action label is annotated at video-level while the other non-action attributes are at frame-level. Nonetheless, PA-HMDB51 only contains about 500 videos, it is only used for evaluation. There are 51 different action classes. The other 5 non-action attributes are all binary. As there is no training data for the non-action attributes in PA-HDMB51, we used VISPR (Orekondy et al., 2017) for training them, and used HMDB51 for the action attribute. MaSS was evaluated on both VISPR and PA-HMDB51.

**Training and Evaluation Protocols.** Before we can start training MaSS' data modifier $G$, we first need to train the models for the attributes we intend to suppress in the suppression branch, as well as the attribute-agnostic model for the preservation branch, and potentially also the attribute-specific models if the corresponding labels for those attributes are available. The training details for those pretrained models are in Appx. B.1 and B.2. Subsequently, each of these pre-trained models is used in either the suppression or the preservation branch during the training of the data modifier $G$ via optimizing Eq. 9 defined in Sec. 3. The details of training MaSS are described in Appx. B.3, including the similarity measurements, hyperparameters, loss weights, etc. Note that during training $G$, the pre-trained models in both branches are fixed without any update. For evaluation, we use the trained data modifier to generate new data and feed them through the pre-trained models and examine their performance. For both Adience and AudioMNIST, we use top-1 accuracy as the metric and report the results on each attribute. For PA-HMDB51, we follow the same practice as PA-HMDB51 to aggregate the performance of the 5 non-action attributes via macro-average of classwise mean average precision (cMAP) and use top-1 accuracy for action.

## 4.2 COMPARISON TO BASELINES

First, we examine the effects of each of the loss terms for each of the datasets. Table 2 shows the performance on Adience where DataID is targeted for suppresion, and the others are expected to be preserved. When only imposing attribute suppression ($L_{\text{rec}}$ and $L^{s_1}$), MaSS did successfully degrade the performance of DataID. But at the same time, the performance on age and gender deteriorated significantly. When we added the attribute-agnostic loss ($L^{r_*}$), MaSS still achieved good suppression on DataID. But at the same time it greatly improved the recognition accuracy on age and gender. Note that, we do not use any information from age and gender attributes during the training. Moreover, when the age information ($L^{r_1}$) was incorporated in MaSS, the performance of age detection on the transoformed dataset almost returned to the same level as that of the original dataset. The same trend can be observed for the gender attribute ($L^{r_2}$) as well.

A similar outcome can also be observed for the AudioMNIST dataset (Table 3), where SpeakerID was treated as the suppression target, and all the rest of the attributes were expected to be preserved. When MaSS only used suppression ($L_{\text{rec}}$ and $L^{s_1}$), SpeakerID score did reduce, but so did the other attributes. Adding the attribute-agnostic loss ($L^{r_*}$) brought significant improvement for the

Table 3: Results on AudioMNIST under different configurations. The checkmark (✓) denotes that the particular loss term is used in the optimization. SpeakerID is selected as the suppression target.

| | Loss Configuration | | | | | | Top-1 Accuracy (%) | | | |
|---|---|---|---|---|---|---|---|---|---|---|
| | $L_{rec}$ | $L^{s_1}$ | $L^{r_*}$ | $L^{r_1}$ | $L^{r_2}$ | $L^{r_3}$ | SpeakerID ($s_1$) | Digit ($r_1$) | Accent ($r_2$) | Gender ($r_3$) |
| Original | - | - | - | - | - | - | 95.6 | 99.8 | 99.3 | 96.5 |
| MaSS | ✓ | ✓ | - | - | - | - | 0.0 | 26.2 | 45.3 | 53.5 |
| | ✓ | ✓ | ✓ | - | - | - | 1.7 | 67.4 | 68.7 | 88.2 |
| | ✓ | ✓ | ✓ | ✓ | - | - | 1.7 | 99.7 | 68.4 | 80.0 |
| | ✓ | ✓ | ✓ | ✓ | ✓ | - | 1.7 | 99.7 | 95.1 | 86.6 |
| | ✓ | ✓ | ✓ | ✓ | ✓ | ✓ | 1.7 | 99.6 | 95.7 | 98.4 |

Table 4: Results on VISPR and PA-HMDB under different configurations. The checkmark (✓) denotes that the particular loss term is used in the optimization. Metrics for the action attribute is Top-1 Accuracy (%) while cMAP (%) is used for the other 5 non-action attributes. MaSS is configured to suppress the non-action attributes.

| | Loss Configuration | | | | VISPR | PA-HMDB | |
|---|---|---|---|---|---|---|---|
| | $L_{rec}$ | $L^{s_1}$ | $L^{r_*}$ | $L^{r_1}$ | Non-action Attrs. ($s_1$) | Action ($r_1$) | Non-action Attrs. ($s_1$) |
| Original | - | - | - | - | 81.8 | 58.7 | 79.7 |
| MaSS | ✓ | ✓ | - | - | 41.8 | 12.6 | 70.3 |
| | ✓ | ✓ | ✓ | - | 36.3 | 52.1 | 63.2 |
| | ✓ | ✓ | ✓ | ✓ | 38.6 | 58.0 | 63.4 |

performance of all the other three attributes. Finally, when the attribute information were available ($L^{r_1}$, $L^{r_2}$ and $L^{r_3}$), MaSS was able to boost the preservation performance to near perfection without reducing the suppression quality on SpeakerID.

Table 4 shows the performance of MaSS on the video dataset. With only suppression ($L_{rec}$ and $L^{s_1}$), MaSS indeed lowered the cMAP on both VISPR and PA-HMDB. However, the performance on the action attribute also dropped significantly. By adding the attribute-agnostic loss ($L^{r_*}$), MaSS was able to significantly improve the accuracy of the action attribute without utilizing any label information. We do notice a slight decrease in the performance of other attributes, which also aligns with the observation in SPAct (Dave et al., 2022), When the action label ($L^{r_1}$) was incorporated, MaSS was able to preserve the same action accuracy while suppressing the other attributes.

All experimental results above show that MaSS is highly configurable and effective even without knowing the downstream tasks in advance. Moreover, when the information of downstream task is available, MaSS can boost the performance on the corresponding specific attributes.

### 4.3 COMPARISON TO OTHER METHODS

As MaSS includes both suppression and preservation, there is limited prior work for direct comparison since most of them focus on suppression only. Therefore, we first compared MaSS with the heuristic methods, e.g., perturbation on the original data, additive noise in the feature space for both the Adience and AudioMNIST datasets. For Adience, we further compared MaSS to CIA-GAN (Maximov et al., 2020), DeepPrivacy (Hukkelås et al., 2019) and Fawkes (Shan et al., 2020) even though they only performed suppression. Lastly, we compared MaSS to SPAct on PA-HMDB. More details on the comparisons can be found in Appx. D.

Table 5 shows the comparison on Adience. As most methods were designed to suppress DataID, we configured MaSS to suppress DataID and preserve the others with attribute-agnostic setting for comparison. First, adding noise in the feature domain lowered the accuracy on the DataID attribute but it also degraded the performance on age and gender. On the other hand, for all the methods which manipulate data in the original domain, including Gaussian blurring with different kernel sizes and standard deviations, downsampling and upsampling back to the original size, and obfuscating the various face area where the face is detected by MTCNN (Schroff et al., 2015), they led to similar results since they did not learn what features should be preserved for downstream tasks. By modifying faces in the image space, CIAGAN (Maximov et al., 2020), DeepPrivacy (Hukkelås

Table 5: Comparison to other methods on Adience. Top-1 accuracy (%) is reported. Features and Raw Data denote that the changes are made on the feature level and the image level, respectively.

| | Method | DataID ($\downarrow$) | Age ($\uparrow$) | Gender ($\uparrow$) |
|---|---|---|---|---|
| Gaussian Noise | Features, $\sigma = 0.5$ | 50.8 | 56.8 | 88.1 |
| | Features, $\sigma = 1.0$ | 0.5 | 26.8 | 54.9 |
| Guassian Blur | Raw Data, $k = 11$, $\sigma = 10.0$ | 39.4 | 49.0 | 86.5 |
| | Raw Data, $k = 21$, $\sigma = 15.0$ | 1.0 | 19.3 | 66.2 |
| Downsample | Raw Data, 8× | 11.0 | 30.3 | 78.7 |
| | Raw Data, 4× | 79.3 | 74.7 | 94.3 |
| Obfuscation | Raw Data, Face area, ratio = 1.0 | 0.8 | 17.1 | 59.5 |
| | Raw Data, Face area, ratio = 0.36 | 4.0 | 29.8 | 81.6 |
| | Raw Data, Face area, ratio = 0.09 | 57.7 | 64.1 | 93.9 |
| CIAGAN | | 1.1 | 17.8 | 66.9 |
| DeepPrivacy | | 5.9 | 32.2 | 84.0 |
| Fawkes | | 24.7 | 47.6 | 87.8 |
| MaSS | $L_{rec} + L^{s_1} + L^{r_*}$ | 0.6 | 78.5 | 95.7 |

Table 6: Comparison with other methods on AudioMNIST. Top-1 accuracy (%) is reported.

| Method | SpeakerID ($\downarrow$) | Digit ($\uparrow$) | Accent ($\uparrow$) | Gender ($\uparrow$) |
|---|---|---|---|---|
| White Noise (Raw Data) | 4.7 | 23.3 | 23.4 | 32.2 |
| Masking (Raw Data) | 1.8 | 10.6 | 3.9 | 80.0 |
| MaSS ($L_{rec} + L^{s_1} + L^{r_*}$) | 1.7 | 67.4 | 68.7 | 88.2 |

et al., 2019) and Fawkes (Shan et al., 2020) were able to lower the performance on DataID again. However, they still weren't capable of preserving any features for the other attributes. In contrast, MaSS not only suppressed DataID but also preserved much more data utility such that the data could still perform well on age and gender detection tasks, even though no additional information on age or gender was made available to MaSS.

The comparative study results on AudioMNIST are listed in Table 6. The heuristic approaches were capable of suppressing SpeakerID, as well as the other attributes indiscriminatively. Nonetheless, MaSS can retain the other attributes without knowing the downstream tasks while still achieving SpeakerID suppression. Lastly, comparing to SPAct (Dave et al., 2022) on PA-HMDB[1], MaSS achieved competitive suppression ratio of cMAP on both datasets (VISPR: 55.6% vs. 57% and PA-HMDB: 20.7% vs. 16%) for the other 5 non-action attributes. In addition, MaSS also provides the flexibility to configure which attributes to suppress while SPAct can only target all five attributes at the same time.

Additional ablation studies of MaSS, including suppressing different/multiple attributes, effects of loss weights and different similarity measurements, can be found in Appx. C.

## 5 CONCLUSION

In this paper, we proposed MaSS to selectively suppress attributes while preserving other attributes; moreover, with the proposed attribute-agnostic approach, the preservation can be achieved without foreseeing the downstream tasks, which expands the usability of the proposed algorithm. We validated our method in three datasets in different domains, including facial images, voice audio and video clips, and all results showed that our method is promising in attribute suppression and preservation. Lastly, we would like to point out that we validated MaSS in the feature space rather than the original data space; an interesting future direction, which is beyond the scope of this paper, would be to integrate MaSS with generative models, such as GAN or diffusion models to further convert the transformed feature vectors back into their original representation space (image, audio, etc.).

---

[1]We compared SPAct based on their Table 8 in the paper, which is the closest to our setting. Because the baseline performance is different, we show here the relative suppression results. See Table 16 for details.

**Code of Ethics and Ethics statement.** Our work is to selectively suppress the attributes in multi-attribute data while preserving their potential utility such that the sensitive information in the data could be minimized and hence the data could be used to benefit the community. We believe that there are no ethical concerns related to this work.

**Reproducibility Statement.** We provided the training and evaluation details in the main paper and appendix. Our source codes and models will be publicly available to help better understand the settings of training and evaluation.

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

APPENDIX

In the appendix, we provided the supplemented materials, including data preprocessing in Sec. A. In Sec. B, we described model pre-training for contrastive learning, model training for single attribute and MaSS. We included the ablation studies in Sec. C and how to generate compared results in Table 5 and Table 6 in Sec. D. Our source code and models will be publicly available to help better understand the settings of training and evaluation.

## A  DATA PREPROCESSING

As briefly discussed in the main manuscript, MaSS takes a feature vector in and then generates a feature vector instead of operating on the raw data. Therefore, for each dataset, we convert the data into feature vectors via state-of-the-art neural networks and then normalize the vector by its L2-norm.

**Adience.**  We first resize the image into $160 \times 160$ and normalize the image by the mean and the standard deviation used in the FaceNet (Schroff et al., 2015). Then, we feed the normalized image into FaceNet to get a 512-d feature vector.

**AudioMNIST.**  The majority of the information in audio signal resides at the beginning, and the average length of a waveform is 30,844 samples and the upper quartile is 34,380. Therefore, we either truncate and pad (zeros) the waveform to the length of 30,000 at the end such that the data loader can form them as a batch to speed up the training. Then, we feed the truncated/padded waveform into Hubert-L (Hsu et al., 2021) to get a 1024-d feature vector after performing average pooling on the output of Hubert-L along the time dimension.

**VISPR and PA-HMDB.**  R3D-18 is trained with the clip size of $16 \times 112 \times 112$ and generates a 512-d feature vector; therefore, we resize the spatial dimension of a video into $112 \times 112$ and then sample 16 frames (every other frame) out of a video to form a clip. For VISPR, since it is an image dataset, we generate a 16-frame clip by duplicating the same image and then pass it to R3D-18 to extract features. On the other hand, for the action attribute in PA-HMDB, we convert each video into frame-level feature vectors by R3D-18. More specifically, for each timestamp, we take its neighboring frame (every other frames) to form a 16-frame clip and then feed it to R3D-18. E.g., for a video with 100 frames, we will get 100 512-d feature vectors. For the other attributes in PA-HMDB, since those labels are image-level instead of video-level, we simply assign the labels to the frame-level features extracted above.

## B  MODEL TRAINING

### B.1  ATTRIBUTE-AGNOSTIC MODEL PRE-TRAINING

For all datasets, we follow similar practices to train attribute-agnostic models via SimCLR (Chen et al., 2020). First, we generate two views of data by different data augmentation in the raw data domain, and then the two views of data are passed through the fixed feature extractor to get its feature representation. After that, we train a multi-layer perceptron (MLP) as an encoder to learn a generic feature representation over the features extracted by the fixed feature extractor. The MLP is composed of two fully-connected layers with the same dimension as the input feature dimension, and 1-D batch normalization layer and ReLU are added between fully-connected layers. The trained MLP is served as attribute-agnostic model in MaSS. Note that we also adopt the MLP-projector in the contrastive learning to achieve better performance. We train the model for 100 epochs with temperature 0.07 via the stochastic gradient decent (SGD) optimizer. The weight decay is set to 0.0001 and the learning rate starts from 0.05 and then it is annealed with cosine schedule. In the follow paragraphs, we describe how to generate different views for each dataset.

**Adience.**  To generate two views, we first resize images to $160 \times 160$ and then randomly flip the image horizontally; after that, we randomly perform color jitter via torchvision package with 80% probability and then convert the image into gray scale with 20% probability.

Table 7: Model training settings for each dataset in Table 2, 3 and 4.

| Dataset | #GPUs | Batch Size per GPU | Learning Rate | Loss Weights |
|---------|-------|--------------------|---------------|--------------|
| Adience | 1 | 64 | 0.00125 | $w_{\text{rec}} = 1, w^{s_1} = 5, h^{s_1} = 1$ 
 $w^{r_*} = 40, w^{r_1} = 1, w^{r_2} = 1$ |
| AudioMNIST | 4 | 128 | 0.01 | $w_{\text{rec}} = 10, w^{s_1} = 0.001, h^{s_1} = 0.1$ 
 $w^{r_*} = 15, w^{r_1} = 0.01, w^{r_2} = 0.001, w^{r_3} = 0.001$ |
| PA-HMDB | 4 | 128 | 0.01 | $w_{\text{rec}} = 1, w^{s_1} = 1, h^{s_1} = 0.1$ 
 $w^{r_*} = 1000, w^{r_1} = 50$ |

Table 8: Results on Adience with different suppressed attribute, the experiments are completed under the loss setting of $L_{rec} + L^{s_1} + L^{r_*}$, where $s_1$ is the suppressed attribute.

| | Suppressed Attribute | Top-1 Accuracy (%) | | |
|---|----------------------|--------|-----|--------|
| | | DataID | Age | Gender |
| MaSS | DataID | 0.6 | 78.5 | 95.7 |
| | Age | 81.2 | 13.7 | 94.5 |
| | Gender | 77.2 | 75.3 | 8.3 |

**AudioMNIST.** In this work, we apply two different augmentations (Ma, 2019): random crop on the entire audio with a coverage of 0.4 and mask with a coverage of 0.5 to each view respectively. The crop augmentation removes the selected part from the audio, whereas mask substitutes it with zeros.

We limited the data augmentations used in our methods to crop and mask because other augmentation like pitch, loudness, speed, etc. would affect the structure of audio signal and potentially won't be able to retain attributes like gender, accent, and age.

**PA-HMDB.** We generate two views of data by following the practice in CVRL (Qian et al., 2021), i.e., for a positive pair, two views are extracted from different time instance of a video and the temporal-consistent data augmentation is performed on each view. The data augmentation is composed of resizing the spatial dimension into $112 \times 112$, guassian blurring, randomly converting color image into gray image.

### B.2 ATTRIBUTE-SPECIFIC MODEL PRE-TRAINING

For all attributes in all datasets, we train the attribute-specific model by using the cross-entropy loss against the given label to compute the gradient for all parameters through back-propagation. The model contains three fully-connected layers and with the dimension: *input dim-512-256-number of classes*, and the 1-D batch normalization layer and ReLU are added between layers. We use a batch size of 256 with the AdamW optimizer (Loshchilov & Hutter, 2019) to train the model for 100 epochs. The weight-decay is fixed to 0.05 and the initial learning rate is set to 0.01 and then the learning rate is annealed with the cosine scheduler.

### B.3 MASS TRAINING

The training on different datasets follows similar settings but with different loss weights. When training the data modifier $G$ in MaSS, all models in the suppression and preservation branches are fixed without any update. We train all models with 100 epochs with the AdamW optimizer (Loshchilov & Hutter, 2019) The weight-decay is 0.05 and we adopt cosine learning scheduler to anneal the learning rate. For the loss type, in most of cases, we use cosine similarity measurement for the to-be-suppressed attribute and KL divergence for the attribute-specific preservation. Table 7 described other training details for different datasets.

Table 9: Results on Adience with different weights on $L^{r_*}$, the experiments are completed under the loss setting of $L_{rec} + L^{s_1} + L^{r_*}$. We used $w^{r_*} = 40$ in our main results.

| | $w^{r_*}$ | Top-1 Accuracy (%) | | |
| | | DataID | Age | Gender |
|---|---|---|---|---|
| | 10 | 0.0 | 69.4 | 73.5 |
| | 20 | 0.0 | 75.0 | 94.8 |
| MaSS | 40 | 0.6 | 78.5 | 95.7 |
| | 80 | 3.2 | 82.1 | 96.4 |
| | 160 | 13.5 | 83.5 | 96.4 |

Table 10: Results on Adience with different similarity measurements in $L^{s_1}$, the experiments are completed under the loss setting of $L_{rec} + L^{s_1} + L^{r_*}$. $s_1$ is DataID.

| | Similarity | Top-1 Accuracy (%) | | |
| | | DataID | Age | Gender |
|---|---|---|---|---|
| | Cosine | 0.6 | 78.5 | 95.7 |
| MaSS | KL divergence | 0.2 | 76.2 | 95.8 |
| | CE | 0.1 | 76.7 | 94.9 |

Table 11: Results on Adience under multiple suppressed attributes. The DataID and gender are selected to be suppressed. MaSS is configured with $L^{r_*}$ and $L^{r_1}$.

| | Top-1 Accuracy (%) | | |
| | DataID ($s_1$) | Gender ($s_2$) | Age ($r_1$) |
|---|---|---|---|
| Original | 90.8 | 97.4 | 89.1 |
| MaSS | 0.9 | 5.0 | 83.8 |

## C  ABLATION STUDIES

In ablation studies, we use the Adience dataset for all experiments and we discuss MaSS in three perspectives, including suppressing different attributes, effects of loss weights, effects of similarity measurement.

**Suppression Target.**  In the main manuscript, we always suppress DataID in all experiments; however, MaSS is configurable to suppress any attribute while still preserving others. Table 8 shows the results by suppressing different attributes. Only the performance of the selected attribute is degraded while other attributes are still good. Note that those results do not include any attribute-specific models in the preservation branch. The result shows that MaSS is flexible to configure to suppress any attribute and preserve others.

**Loss Weights.**  Intuitively, the loss weight controls which loss term should be focused on more during the optimization. In this ablation study, we vary the weights for the attribute-agnostic model and the results are shown in Table 9. Since $L^{r_*}$ controls how generic the feature representation is, the higher weights preserve more generic features; therefore, the transformed dataset could perform better for all attributes. However, when $L^{r_*}$ is 160, the accuracy of DataID is also increased because the strength of suppression is not strong enough since the weight of $L^{r_*}$ is too high.

**Different Similarity Measurement for Suppression.**  We proposed three different measurements for the similarity in the main manuscript. Those measurements provided similar functionalities conceptually but they might work different empirically. Table 10 shows the results with different measurements, and all results are close to each other. Therefore, for suppression, we use cosine for all experiments.

**Multiple Suppressed Attributes.**  Table 11 shows the results when we configured MaSS to suppress multiple attributes, DataID and gender. The accuracy of both DataID and gender attributes are

Table 12: Ablation studies of losses in the suppression branch on Adience. DataID is selected as the suppression target.

| | Loss Configuration | | | | | Top-1 Accuracy (%) | | |
| --- | --- | --- | --- | --- | --- | --- | --- | --- |
| | $L^{s_1}$ | $L_{rec}$ | Entropy Loss | L2 Loss | Entropy | DataID ($s_1$) | Gender | Age |
| MaSS | ✓ | - | - | 0.003 | 5.4 | 0.0 | 28.6 | 68.9 |
| | ✓ | ✓ | - | 0.002 | 5.4 | 0.0 | 27.4 | 68.1 |
| | ✓ | ✓ | ✓ | 0.002 | 6.6 | 0.0 | 33.8 | 73.5 |

Table 13: Results on Adience under different configurations. DataID is selected as the suppression target.

| | Loss Configuration | | | | | Top-1 Accuracy (%) | | |
| --- | --- | --- | --- | --- | --- | --- | --- | --- |
| | $L_{rec}$ | $L^{s_1}$ | $L^{r_*}$ | $L^{r_1}$ | $L^{r_2}$ | DataID ($s_1$) | Age ($r_1$) | Gender ($r_2$) |
| Original | - | - | - | - | - | 90.8 | 89.1 | 97.4 |
| MaSS | ✓ | ✓ | ✓ | ✓ | ✓ | 0.6 | 86.9 | 96.7 |
| | ✓ | ✓ | - | ✓ | ✓ | 0.0 | 84.5 | 96.3 |

Table 14: Results on AudioMNIST under different configurations. The checkmark (✓) denotes that the particular loss term is used in the optimization. SpeakerID is selected as the suppression target.

| | Loss Configuration | | | | | | Top-1 Accuracy (%) | | | |
| --- | --- | --- | --- | --- | --- | --- | --- | --- | --- | --- |
| | $L_{rec}$ | $L^{s_1}$ | $L^{r_*}$ | $L^{r_1}$ | $L^{r_2}$ | $L^{r_3}$ | SpeakerID ($s_1$) | Digit ($r_1$) | Accent ($r_2$) | Gender ($r_3$) |
| Original | - | - | - | - | - | - | 95.6 | 99.8 | 99.3 | 96.5 |
| MaSS | ✓ | ✓ | ✓ | ✓ | ✓ | ✓ | 1.7 | 99.6 | 95.7 | 98.4 |
| | ✓ | ✓ | - | ✓ | ✓ | ✓ | 1.6 | 99.7 | 95.0 | 97.7 |

Table 15: Results on VISPR and PA-HMDB under different configurations. The checkmark (✓) denotes that the particular loss term is used in the optimization. Metrics for the action attribute is Top-1 Accuracy (%) while cMAP (%) is used for the other 5 non-action attributes. MaSS is configured to suppress the non-action attributes.

| | Loss Configuration | | | | VISPR | PA-HMDB | |
| --- | --- | --- | --- | --- | --- | --- | --- |
| | $L_{rec}$ | $L^{s_1}$ | $L^{r_*}$ | $L^{r_1}$ | Non-action Attrs. ($s_1$) | Action ($r_1$) | Non-action Attrs. ($s_1$) |
| Original | - | - | - | - | 81.8 | 58.7 | 79.7 |
| MaSS | ✓ | ✓ | ✓ | ✓ | 38.6 | 58.0 | 63.4 |
| | ✓ | ✓ | - | ✓ | 38.3 | 52.2 | 63.6 |

suppressed successfully but the age attribute can still be recognized even without age labels. With age label, MaSS can further improve its accuracy.

**Losses in Suppression Branch.**  Here, we further explore the effects of L2 reconstruction loss and prediction entropy loss applied in the suppression branch.

We add the L2 reconstruction loss to minimize changes in the data while suppressing the targeted attributes; thus, we are able to keep the data as truthful as possible. The prediction entropy loss is added to increase the entropy of the predicted probability such that the prediction of the modified data becomes closer to a random guess; thus, there is less information in the prediction.

Table 12 shows the effects of these two loss terms when only suppression is considered, i.e., no preservation branch. With L2 reconstruction loss, the overall performance of attribute recognition does not affected; however, it reduced the overall L2 loss. After adding prediction entropy loss, the accuracy of DataID is still 0% but its entropy is increased, which makes the prediction closer to a random guess, as desired.

Table 16: Comparison to SPAct (Dave et al., 2022), Metrics for the action attribute is Top-1 Accuracy (%) while cMAP (%) is used for the other 5 non-action attributes. Non-action attributes are suppressed in the experiment, and MaSS is configured without $L^{r_1}$.

| | VISPR
Non-action Attrs. ($s_1$) | PA-HMDB
Action ($r_1$) | Non-action Attrs. ($s_1$) |
|---|---|---|---|
| Original | 81.8 | 58.7 | 79.7 |
| MaSS | 36.3 ($\downarrow$55.6%) | 52.1 | 63.2 ($\downarrow$20.7%) |
| Original (SPAct) | 64.4 | - | 70.1 |
| SPAct | 27.4 ($\downarrow$57%) | - | 58.9 ($\downarrow$16%) |

**Effects of $L^r_*$ with All Labels.** In Table 2, 3 and 4, we have shown that the benefits to have attribute-agnostic loss ($L^{r_*}$) compared to suppression only. Here, we discussed the contribution of $L^{r_*}$ when all to-be-preserved attributes are available, and the results are shown in Table 13, 14 and 15. Without $L^{r_*}$, it achieved competitive performance on to-be-preserved attributes to the one with $L^{r_*}$ on Adience and AudioMNIST for all attributes. Moreover, for PA-HMDB, without $L^{r_*}$, the accuracy of action is degraded 5.8%, we conjecture that even though the generic feature retained by enforcing $L^{r_*}$ help the data utilities of the modified data.

# D    COMPARED RESULTS

**Adience.** We compared many approaches in Table 5 and here we describe the details for how to generate those results. First, for Gaussian noise, we added zero-mean with different standard deviations ($\sigma$) into the original feature vectors to manipulate data. For Gaussian blur, downsample and obfuscation are all performed in the raw data domain, and then the modified data are passed through FaceNet to get the feature representation. For Gaussian blur, we apply zero-mean with various standard deviations ($\sigma$) with different kernel sizes ($k$) to blur the image. For downsample, we downsample the data with different ratios and then upsample it back to original size. Lastly, for obfuscation, we use MTCNN to detect the location of the face and then remove the face region with different ratios.

For CIAGAN (Maximov et al., 2020), we first followed CIAGAN's method to pre-extract the masked face and the facial landmark information for the Adience dataset by using the Dlib-ml library (King, 2009). And then, the CIAGAN model takes in the Adience images, their landmarks, masked faces and the desired target. For DeepPrivacy (Hukkelås et al., 2019) and Fawkes (Shan et al., 2020), we adjusted the released codes and ran over the Adience dataset to generate new images.

After we obtain the transformed Adience images, we use the same procedure as ours for evaluation: using FaceNet (Schroff et al., 2015) to extract the feature vector of an image.

**AudioMNIST.** We compared two methods in AudioMNIST, including adding white noise and masking out a portion of waveform based on the nlpaug library (Ma, 2019). We use the default parameter for white noise and set the masking ratio to 50% of the waveform.

**SPAct.** We tried our best to compare with SPAct (Dave et al., 2022) under the same experimental condition; however, our absolute performance over the raw data is significantly better than their paper; therefore, we only compare the relative gains to them. Moreover, under this setting, they do not show the accuracy of the action attribute. Table 16 shows the comparison to SPAct and baselines. When comparing to its own baseline, our method is competitive in suppressing non-action attributes while keeping good accuracy in the action attribute.

