# OpenReview forum: "MaSS: Multi-attribute Selective Suppression"
_ICLR.cc/2023/Conference — Submitted to ICLR 2023_

### Official Review · Reviewer_uwsC · 2022-10-18

**Confidence:** 2
**Correctness:** 3
**Technical Novelty And Significance:** 2
**Empirical Novelty And Significance:** 3
**Recommendation:** 6

**Clarity, Quality, Novelty And Reproducibility:**

The paper is generally written clearly. I am not an expert in this area and cannot evaluate the originality of this work in great detail. The method makes sense to me.

**Strength And Weaknesses:**

strength:
1. the paper is easy to follow and written well.
2. the motivation of this paper is valid and important to me, i.e., suppressing the data properly without affecting other attributes.
3. the results seem to be effective from the main tables on main benchmarks.

weakness:
1. in section 3.2.2, the author mention that "It utilizes the corresponding set of inference models pretrained on the original dataset X". It is not clear how the pretrained models are trained such that it only focuses on one specific attribute.
2. in section 3.2.3, for the agnostic case, it is not clear why a specific self-supervised pretraining can help preserve the non-selected attributes. If the selected attributes are forced to have a similar embedding after transformation, isn't it harmful to the final goal?
3. the comparison in the experiment section is not consistent, for example, the author compares with the SPAct in PA-HMDB dataset, but they did not compare with this method on the ADIENCE dataset, which is their main result.


**Summary Of The Paper:**

This paper proposes Multi-attribute Selective Suppression, or MaSS, a general framework for performing precisely targeted data surgery to simultaneously suppress any selected set of attributes while preserving the rest for downstream machine learning tasks. The end goal is to achieve a sophisticated mechanism that gives data owners fine-grained control while retaining the maximal degree of data utility.

**Summary Of The Review:**

The paper utilizes an adversarial training-based learning algorithm to fulfill their goal-simultaneously suppressing any selected set of attributes while preserving the rest for downstream machine learning tasks. There are some vague explanations in the paper that need to be clarified. Will consider changing my score after I see some of the other reviewers' opinions and the authors' responses.

---

> ### Author Response · Authors · 2022-11-19
> **Response to Reviewer uwsC**
>
> We thank the reviewer for the recognition of the paper as being well-written, the motivation as being valid and important, and the results as effective. We addressed the weaknesses raised by the reviewer, as follows.
>
>
> **[W1] It is not clear how the pretrained models are trained such that it only focuses on one specific attribute.**
>
> All the attribute-specific pretrained models in both the suppression and preservation branches are trained separately with single attribute using supervised learning.
> As the labels of attributes are given, we trained those models by standard cross-entropy loss. The details of are described in Appx. B.2.
> On the other hand, for attribute-agnostic pretrained models, we trained the models using SimCLR to obtain a model that could extract generic features. The details are in Appx. B.1.
> Furthermore, all our code and models will be released.
>
>
> **[W2] it is not clear why a specific self-supervised pretraining can help preserve the non-selected attributes.**
>
> This is one of the interesting findings in this work.
> Self-supervised learning has been widely-studied to provide the generic feature representation without label information and the extracted features usually work well with many downstream tasks.
> Thus, based on this observation, we proposed to use self-supervised pretrained models to preserve the non-selected attributes through retaining generic features in the data.
> Then, we empirically validated our intuition with three datasets and all results show that the self-supervised learning improves the performance of the modified data on downstream tasks significantly without utilizing the labels of those tasks. (As shown in Table 2, 3 and 4 in the latest manuscript.)
>
>
> **[W3] the comparison in the experiment section is not consistent...**
>
> We would like to observe that most papers only try to suppress the attribute instead of doing both suppression and preservation. Moreover, those methods are usually only applicable to specific types of data.
> E.g., CIAGAN is designed for the face dataset (Adience); while SPAct is crafted for the action dataset (PA-HMDB).
>
> In our revised manuscript, we added comparative experiments of two SOTA methods, DeepPrivacy [1] and Fawkes [2], on the Adience dataset. As shown by the results in the table below, in suppressing the DataID attribute, these two methods also cause significant degradation to the recognition of the age and gender attributes.
> The new results are added in Table 5 of the latest manuscript.
>
> **Table: More comparison to other methods on Adience. DataID is selected as the suppression target. Top-1 Accuracy (\%) is reported.**
>
> |  | DataID |  Age | Gender |
> | --- | --- | --- | --- |
> | Original | 90.8 | 89.1 | 97.4 |
> | CIAGAN |  1.1 | 17.8 | 66.9 |
> | DeepPrivacy |  5.9 | 32.2 | 84.0 |
> | Fawkes | 24.7 | 47.6 | 87.8  |
> | Ours |  0.6 | 78.5 | 95.7 |
>
>
> Reference:
>
> [1] Håkon Hukkelås, Rudolf Mester, Frank Lindseth, "DeepPrivacy: A Generative Adversarial Network for Face Anonymization," ISVC 2019.
>
> [2] Shan, Shawn, et al. "Fawkes: Protecting privacy against unauthorized deep learning models." 29th USENIX security symposium (USENIX Security 20). 2020.

---

### Official Review · Reviewer_SL3b · 2022-10-24

**Confidence:** 4
**Correctness:** 3
**Technical Novelty And Significance:** 3
**Empirical Novelty And Significance:** 3
**Recommendation:** 5

**Clarity, Quality, Novelty And Reproducibility:**

The clarity of the paper can be improved (see the first and second points in Cons).

The novelty of the proposed method is moderate. It notices the drawbacks of previous methods (e.g. not preserving useful information, not flexible enough) and addresses them, but the loss function just combines several natural and standard terms.

The proposed method should be reproducible since most of the details are provided.

**Strength And Weaknesses:**

Pros:
- The topic is interesting and the motivation is strong. Removing sensitive attributes from data is crucial for real-world data usage, and (as shown in Table 5) vanilla or prior data obfuscation methods either cannot successfully destroy the performance on targeted attributes or cannot preserve satisfiable performance on other attributes. This paper aims at reaching the two goals simultaneously.
- The proposed method is straightforward, flexible and effective. Each components of the proposed loss function directly relates to a desired purpose. The proposed method can suppress an arbitrary set of attributes and can preserve information in both attribute-agnostic and attribute-specific situation. Experiments show the proposed method can decrease the performance on targeted attributes drastically with little impact on the preserved attributes.

Cons:
- The abstract is too long. It should be more concise.
- The method section and Figure 2 are a little bit confusing. It is not clear how many pre-trained models are used and where they are. It is neither clear which loss is penalty and which loss is award. For example, both the penalty and the award similarity loss for the two branches are called "Attribute-Specific Similarity loss", which is very confusing.  Also, why is there only one Prediction Entropy Loss in the Suppression Branch in Figure 2? Shouldn't it be one for each attributes and thus S in total? Besides, what does it mean by "the data modifier tries to minimax the loss"? Isn't the data modifier the only updating network which is minimizing the loss while others are pre-trained and fixed models?
- All data points are converted into feature embeddings in the experiments. It not only requires a good feature extractor in advance when applying the proposed method, but also prevent the usage of domain-specific techniques (e.g. data augmentations, neural architectures) when data users uses the processed data.
- The ablation of the proposed loss function is not enough. Since the proposed loss function is composed of so many terms, it will be considered as validated only if each component shows effect. Currently, there's no experiments demonstrating the benefit from the reconstruction loss. For the suppression branch, there's no experiments showing how many benefits come from the Attribute-Specific Similarity loss and how many benefits come from the Prediction Entropy Loss. For the preservation branch, there's no experiments testing whether the Attribute-Agnostic Loss still contributes to preserving the information when the Attribute-Specific Loss is available.

**Summary Of The Paper:**

This paper designs a data obfuscation framework that is able to suppress an arbitrary set of attributes while preserving all other attributes. The proposed framework trains a neural data (additive) modifier to minimize the similarity between the original data and the modified data for the attributes to be suppress while maximizing the similarity between the original data and the modified data for other attributes. The proposed method demonstrates promising results for suppressing targeted attributes while preserving others on multiple datasets.

**Summary Of The Review:**

This paper proposes a data obfuscation framework to selectively suppress attributes while preserving other attributes. The proposed method is straightforward, flexible and effective and the experimental results are great. However, it has some evident drawbacks (e.g. all data points should be converted into feature embeddings) and the paper writing can be improved (e.g. too long abstract, confusing method section, lack of ablation experiments). Also, the proposed method just combines several natural and standard terms, and the ablation study does not convincingly show the efficacy of each (e.g. the reconstruction loss). Therefore, the paper still needs some work.

---

> ### Author Response · Authors · 2022-11-19
> **Response to Reviewer SL3b**
>
>
> We thank the reviewer for recognizing the strength of our work! Our point-by-point clarifications can be found as follows.
>
> **[Con 1] The abstract is too long.**
>
> We have revised our abstract as follows and updated the manuscript accordingly.
>
> > The recent rapid advances in machine learning technologies largely depend on the vast richness
> of data available today.
> Along with the new services and applications enabled by those machine learning models,
> various governmental policies are put in place
> to regulate such data usage and protect people's privacy and rights.
> As a result, data owners often opt for simple data obfuscation
> (e.g., blur people's faces in images)
> or withholding data altogether,
> which leads to severe data quality degradation
> and greatly limits the data's potential utility.
> Aiming for a sophisticated mechanism which gives data owners fine-grained control while retaining the maximal degree of data utility,
> We propose *Multi-attribute Selective Suppression*, or *MaSS*,
> a general framework for performing precisely targeted data surgery
> to simultaneously
> *suppress* any selected set of attributes
> while *preserving* the rest for downstream machine learning tasks.
> MaSS learns a data modifier through adversarial games
> between two sets of networks,
> where one is aimed at suppressing selected attributes,
> and the other ensures the retention of the rest of the attributes.
> We carried out an extensive evaluation of our proposed method
> using multiple datasets from different domains
> including facial images, voice audio, and video clips,
> and obtained highly promising results
> in MaSS' generalizability and capability of drastically suppressing targeted attributes
> while imposing virtually no impact on the data's usability
> in other downstream machine learning tasks.
>
> **[Con 2-1] It is not clear how many pre-trained models are used and where they are.**
>
> Each to-be-suppressed attribute in the suppression branch has one pretrained model and one attribute-agnostic model in the preservation branch.
> If the to-be-preserved attributes are known in advance, they would each have one pretrained model as well.
> All those models in both branches are pretrained separately before the MaSS training and are fixed during MaSS training. Details on how to pretrain those models are illustrated in Appx. B.1 and B.2.
>
> **[Con 2-2] Attribute-Specific Similarity loss in both branches.**
>
> We thank the reviewer for pointing this out. Even though both are attribute-specific similarity losses, the one in the suppression branch **penalizes** the data modifier when the feature/logit of original data is similar to the feature/logit of modified data (minimizing) while the one in the preservation branch **rewards** the similarity (maximizing).
>
>
> **[Con 2-3] Prediction Entropy Loss in the Suppression Branch.**
>
> We thank the reviewer for pointing this out as well. Each to-be-suppressed attribute has its prediction entropy loss. We updated the figures accordingly in the revised manuscript (Figure 2.)
>
> **[Con 2-4] the data modifier tries to minimax the loss.**
>
> We mean the data modifier is trying to minimize the similarity measurement in the suppression branch while maximizing the similarity measurement in the preservation branch.
> In the end, the data modifier minimizes the total loss because of the negative sign before the maximization term.
>
> **[Con 3] All data points are converted into feature embeddings in the experiments.**
>
> In this paper, we explore the idea to suppress selected attributes while retaining its data utility and validated the method under feature embeddings. We agree with the reviewer's concern about exploring the domain-specific techniques. On one hand, we think as large models become the main trend in the ML community, using those models to extract features is a common practice as training those large models is generally not feasible. On the other hand, we are actively exploring the combination with generative models to further convert the data back to the original domain to enable the possibility of exploring domain-specific techniques, as the reviewer suggested.

---

> > ### Author Response · Authors · 2022-11-19
> > **Response to Reviewer SL3b (Cont.)**
> >
> > **[Con 4-1] L2 reconstruction loss and prediction entropy loss in the suppression branch.**
> >
> > We added the L2 reconstruction loss to minimize changes in the data while suppressing the targeted attributes. Thus, we are able to keep the data as truthful as possible. The prediction entropy loss is added to increase the entropy of the predicted probability such that the prediction of the modified data becomes closer to random guess. Thus, there is less information in the prediction.
> >
> > The table below shows the effect of these two loss terms when only suppression is considered. I.e., no preservation branch. The table is also added in the revised manuscript as Table 12.
> > With L2 reconstruction loss, the overall performance of attribute recognition is not affected. However, it reduced the overall L2 loss.
> > After adding prediction entropy loss, the accuracy of DataID is still 0\% but with an elevated entropy, which makes the prediction closer to a random guess, as desired.
> >
> >
> > **Table: Ablation studies on L2 reconstruction loss and prediction entropy Loss on Adience. DataID is selected as the suppression target.**
> >
> > | $L^{s_1}$ | $L_{rec}$ | Entropy Loss | L2 Error | DataID (Entropy) | DataID ($s_1$) (Top-1) | Age (Top-1) | Gender (Top-1) |
> > | ---   | ---  | ---     | ---     | ---    | --- | ---    | --- |
> > | V |  - | - | 0.003 | 5.4 | 0.0  | 28.6 | 68.9 |
> > | V |  V | - | 0.002 | 5.4 | 0.0  | 27.4 | 68.1 |
> > | V |  V | V | 0.002 | 6.6 | 0.0  | 33.8 | 73.5 |
> >
> >
> > **[Con 4-2] Attribute-agnostic loss when all attributes are given in the preservation branch.**
> >
> > First, we would like to emphasize the advantages of using $L^{r_{\*}}$ is that we do not require any label in advance for MaSS to be able to perverse generic features in the modified data. This is desirable in practice as the data owner might not know which attributes are required beforehand. E.g., for the configuration with
> > $L_{rec}+L^{s_1}+L^{r_{\*}}$ in Table 2, 3 and 4 in the paper, MaSS achieved good performance without using the label information of the to-be-preserved attributes.
> >
> > The tables below show the experiments suggested by the reviewer, which are also added in the revised manuscript (Table 13, 14 and 15).
> >
> > **Table: [Adience] Ablation studies with $L^{r_*}$ but all to-be-preserved attributes. DataID is selected as the suppression target. Top-1 Accuracy (\%) is reported.**
> >
> > | | With $L^{r_*}$ | DataID | Age | Gender |
> > | --- | --- | --- | --- | --- |
> > | Ours | Yes | 0.6 | 86.9 | 96.7 |
> > | Ours | No | 0.0 | 84.5 | 96.3 |
> >
> > **Table: [AudioMNIST] Ablation studies with $L^{r_*}$ but all to-be-preserved attributes. SpeakerID is selected as the suppression target. Top-1 Accuracy (\%) is reported.**
> >
> > | | With $L^{r_*}$ | SpeakerID | Digit | Accent | Gender |
> > | --- | --- | --- | --- | --- | --- |
> > | Ours | Yes | 1.7 | 99.6 | 95.7 | 98.4 |
> > | Ours | No | 1.6 | 99.7 | 95.0 | 97.7 |
> >
> > **Table: [PA-HMDB] Ablation studies with $L^{r_*}$ but all to-be-preserved attributes. Non-action attributes are selected as the suppression targets. Top-1 Accuracy (\%) is reported for Action and cMAP (\%) is reported for Non-action.**
> >
> > | | With $L^{r_*}$ | VISPR (Non-action) | PA-HMDB (Action) | PA-HMDB (Non-action) |
> > | --- | --- | --- | --- | --- |
> > | Ours | Yes | 38.6 | 58.0 | 63.4 |
> > | Ours | No | 38.3 | 52.2 | 63.6 |
> >
> > Even without $L^{r_{*}}$, it achieved competitive result on the to-be-preserved attributes for both $L^{r_{\*}}$ on Adience, as well as all attributes on AudioMNIST. We see that for PA-HMDB, without $L^{r_\*}$, the accuracy of the action attribute degraded by 5.8\%. We conjecture that the generic feature retained by $L^{r_\*}$ helps the data utility of the modified data.
> >
> > **[Novelty.]**
> >
> > We thank the reviewer for recognizing that our work resolves the drawbacks of previous methods (e.g., not preserving useful information, not flexible enough). We also explored that using self-supervised learning to preserve generic features on the modified data for better utilities without knowing which attribute(s) to be preserved.

---

> > > ### Comment · Reviewer_SL3b · 2022-11-28
> > > **Follow-up response**
> > >
> > > Thanks for adding the experiments, but I am still concerned about the effect of the reconstruction loss. The experiments shown in the answer of [Con 4-1] suggest that the reconstruction loss is not important empirically.

---

> > > > ### Author Response · Authors · 2022-11-30
> > > > **Authors' Response for Reviewer SL3b**
> > > >
> > > > The L2 reconstruction loss is added to regularize the network to assure the model does not transform x to a completely different x'. Besides the attribute suppression and preservation goals repeatedly reiterated in the paper, another objective of ours is to keep the transformed data as truthful (i.e., close to the original) as possible. Empirically, it might not affect the recognition accuracy for the datasets we are using but it could be potentially useful for other datasets.

---

> > ### Comment · Reviewer_SL3b · 2022-11-28
> > **Follow-up response**
> >
> > Thanks for the detailed response and revision, but [Con 2-1] and [Con 2-2] are misunderstood. I *understood* the points you are trying to clarify, but what I wanted to say is they are *not clear*. For the problem of which parts of the method need pre-trained models ([Con 2-1]), it should be claimed clearly in the method section, and would be better if can be illustrated in the pipeline figure, instead of deferring to
> > "Training and Evaluation Protocols". For the Attribute-Specific Similarity Loss, it should be clearly illustrated in Figure 2. For example, one can use different colors for award and penalty or add a minus sign to avoid confusion if don't want to use different names. Currently, Figure 2 contains too little information and is not self-contained.

---

> > > ### Author Response · Authors · 2022-11-30
> > > **Authors' Response for Reviewer SL3b**
> > >
> > > Thanks for the comments. In our paper, Figure 2 is intended to only provide a high level overview of MaSS, as also suggested by the figure's title. Hence we chose to only show the major functional blocks as opposed to including many details, which we feared might make the figure too busy and add to some readers' confusion early on as opposed to serving its intended purpose of drawing the big conceptual picture. Each big functional block (e.g., Suppression Branch, Preservation Branch, etc) is then described in detail in its corresponding main text section.
> > >
> > > Above said, we do like the reviewer's suggestion of using colors/signs in the figure to improve clarity and will update the figure accordingly.
> > >
> > > In the Proposed Method section, the first paragraph of Section 3.2.2 describes that for each suppressed attribute $s$, an inference model, which is composed of a feature extractor ($F^s$) and a projector ($P^s$), is pretrained on original data **X** with the label of attribute $s$. Since the training of a recognition model is just a standard practice and those models do not require any special training methods, we simply refer the readers to how we trained the models in the other section rather than claiming the training method as part of our own proposed method.
> > > For the preservation branch in Section 3.2.3, the **Attribute-Specific** subsection	then refers to the suppression branch section due to the same procedure in model pretraining. Then, the **Attribute-Agnostic** subsection talks about a model pretrained with SimCLR on the original data and is used to measure the attribute-agnostic similarity.
> > > Therefore, we feel our paper presents a clear description on all pretrained model usages.

---

### Official Review · Reviewer_8b8w · 2022-10-24

**Confidence:** 4
**Clarity, Quality, Novelty And Reproducibility:** 1.Clarity
**Correctness:** 3
**Technical Novelty And Significance:** 2
**Empirical Novelty And Significance:** 3
**Recommendation:** 5

**Details Of Ethics Concerns:**

Author has answered and discussed ethical issues.

**Strength And Weaknesses:**

Advantages:
1. The research topic is helpful for data protection and privacy.
2. The paper tries to verify the proposed method in different domains.
3. The paper elaborates on the research question and background in detail.

Disadvantages:
1. The paper does not describe the details of the method, and it is not easy to reproduce the paper.
2. The paper lacks some very important references and related work.
3. The paper lacks some necessary experiments and analysis.

Questions: 1. Is the role of the DATA MODIFIER to normalize the data? What is the value of n in Equation 1? How to set the residual shortcut here?

2. Does the loss in Equation 2 compute the similarity of each pair of data? Are the same settings used here for all three tasks?

3. How are the attribute-agnostic logits in Equations 5 and 6 estimated?

4. Which formula does $L^{r_1}$ and $L^{r_2}$ in Experiment Table 2 equal? I didn't find it in the paper.

5. The author's method in Table 5 seems to perform similarly to the Downsample. What is the reason for this?

6. Has the author performed other verifications? Does the paper only achieve attribute recognition?

7. Did the author test directly on the original data? Is it using the features already extracted from the dataset for testing?

8. The authors leave out some work on multitask attribute recognition, eg.[1][2]. Could the proposed approach be extended to these multi-attribute recognition methods?

[1] Huang, Siyu, et al. "Gnas: A greedy neural architecture search method for multi-attribute learning." Proceedings of the 26th ACM international conference on Multimedia. 2018.
[2] Cheng, Zhi-Qi, et al. "Learning to transfer: Generalizable attribute learning with multitask neural model search." Proceedings of the 26th ACM international conference on Multimedia. 2018.

**Summary Of The Paper:**

Research Question: Solve problems where simple or complete deletion of data attributes results in severe degradation of data quality. For example, having to lose data for user privacy.

Research Target: The paper proposes a method to suppress any selected attribute in a multi-attribute dataset. The paper highlights that the proposed method can leave the remaining properties unchanged for potential ML-based downstream analysis tasks.

Research Solution: The solution proposed in the paper is to learn data modifiers through an adversarial game between two sets of networks, one designed to suppress selected attributes and the other to ensure the remaining attributes are preserved through a general contrastive loss and explicit classification metrics.

The paper validates the proposed method in three domains: Facial Images, Voice Audio, and Video Clips.

**Summary Of The Review:**

This paper investigates an interesting question and has practical value. But some details of the methodology and experiments confuse me. I would like the author to reply in the rebuttal.

Because of the problems existing in the current paper, my initial score is: marginally above the acceptance threshold.

---

> ### Author Response · Authors · 2022-11-19
> **Response to Reviewer 8b8w**
>
> We thank the reviewer for recognizing that our work is helpful for protection and privacy. We addressed and answered the concerns from the reviewer as follows.
>
> **[W1] The paper does not describe the details of the method and it is not easy to reproduce the paper.**
>
> Our implementation and training details are mostly described in the appendix (Appx. B), e.g., model definition of pretrained models, hyperparameters and loss weights for each dataset. Additionally, our source code and models will be released.
> We modified the experimental setup section of the paper to clarify how to train the pretrained models as well as MaSS itself.
>
> **[W2] The paper lacks some very important references and related work.**
>
> We thank the reviewer for bringing to our attention the additional references. Correspondingly in our revision, we added a paragraph in the related works to discuss multi-attribute learning.
>
> > **Multi-Attribute Learning.**
> Data could contain multiple attributes. For recognition, using one model for each attribute might provide good recognition accuracy but might not scale.
> Multi-attribute learning aims for resolving this problem by learning underlying relationship among attributes such that the model can adaptively handle different attributes [1, 2].
> In this paper, we validated our method with the single-attribute models.
> The multi-attribute model can also be used in our method by replacing multiple single-attribute models.
>
>
> Reference:
>
> [1] Huang, Siyu, et al. ”Gnas: A greedy neural architecture search method for multi-attribute learning.” Proceedings of the 26th ACM international conference on Multimedia. 2018.
>
> [2] Cheng, Zhi-Qi, et al. ”Learning to transfer: Generalizable attribute learning with multitask neural model search.” Proceedings of the 26th ACM international conference on Multimedia. 2018.
>
> **[W3] The paper lacks some necessary experiments and analysis.**
>
> In the revision, we added the following experiments and ablation studies to enhance our experimental evaluation.
>
> 1. Two more comparisons on Adience (Table 5),
> 2. Suppression of multiple attributes (Table 11), and
> 3. More ablation study on the losses (Table 12-16).
>
> **[Q1] Role of data modifier? What is the n? how to set residual shortcut?**
>
> The role of the data modifier is to generate the modified data $\mathbf{x'}$ from the original data $\mathbf{x}$ such that the suppressed attributes can no longer be reliably inferred by machine learning models while other preserved attributes are still discoverable in the modified data $\mathbf{x'}$.
> $n(\cdot)$ is a vector-normalization function, and it is defined right after Eq. 1.
> The residual shortcut is added on top of the normalized output of MLP. In this way, the MLP only needs to learn the delta of the modification.
>
> **[Q2] Loss in Equation 2.**
>
> The loss is computed based on each pair of the original data and the modified data, and then total loss is computed by averaging over a batch of data for each iteration.
> The details of settings on loss weights and simlarity measurements for each dataset can be found in Appx. B and Table 7. We updated the experimental setup section of the paper to clarify these settings.
>
> **[Q3] Attribute-agnostic logits in Equations 5 and 6.**
>
> The logits are computed by feeding the data $\mathbf{x}$ and the modified data $\mathbf{x'}$ to the pretrained attribute-agnostic network ($F^{r_\*}$ and $P^{r_\*}$).
> The pretrained attribute-agnostic network is obtained by training on data without label using SimCLR. The training details can also be found in Appx. B.1.
>
> **[Q4] $L^{r\_1}$ and $L^{r\_2}$ in Table 2.**
>
> $r\_1$ and $r\_2$ are used to refer to the to-be-preserved attributes 1 and 2. In this example, $r\_1$ and $r\_2$ are one instantiation of $r$ in Eq. 8 and $r\_1$ and $r\_2$ belong to set $R$.
>
> **[Q5] Comparison to down-sampling in Table 5.**
>
> Our method is actually better than downsampling. As shown in Table 5, downsampling either does not suppress any attribute (4x) or suppress all attributes (8x). On the other hand, our MaSS suppressed the DataID while keeping good accuracy on the other two attributes: age and gender.
>
> **[Q6] Other verifications? or only attribute recognition.**
>
> In this paper, we measure the performance of the algorithm by attribute recognition as it is widely used to evaluate performance of classification tasks. We envision other measurements, like membership inference attacks [1], could be used as a metric as well.
>
> Reference:
>
> [1] R. Shokri, M. Stronati, C. Song and V. Shmatikov, "Membership Inference Attacks Against Machine Learning Models," 2017 IEEE Symposium on Security and Privacy (SP), 2017.

---

> > ### Author Response · Authors · 2022-11-19
> > **Response to Reviewer 8b8w (Cont.)**
> >
> >
> > **[Q7] Test directly on the original data?**
> >
> > The data modifier generates a modified feature vector from the original feature vector extracted from the original data. Thus, we use the extracted feature vector for testing.  We are actively working on extending our method to the original data space.
> >
> >
> > **[Q8] Multitask attribute recognition.**
> >
> > We thank the reviewer for the suggestion. This is an interesting direction as the multi-attribute models try to build the underlying connections between attributes.
> > We think that the proposed method could be extended to those multi-attribute models by replacing multiple attribute-specific models in the suppression branch to one multi-attribute model.
> > Furthermore, we think multi-attribute models could further benefit our approach by indicating the common features in the to-be-suppressed attributes.

---

### Official Review · Reviewer_HMFn · 2022-10-31

**Confidence:** 3
**Correctness:** 3
**Technical Novelty And Significance:** 3
**Empirical Novelty And Significance:** 2
**Recommendation:** 5

**Clarity, Quality, Novelty And Reproducibility:**

The paper is written in a clear way where most of the illustration, statement and equations are in a clear way with little confusion. Regarding some designing, there are some points to be further discussed.

1. in suppression branch, for the similarity definitions, the authors propose it could be cosine similarity or negative KL-divergence. Firstly, it should be specific enough, i.e., for all the experiments, what exact similarity measure is utilized?

Secondly, consider the suppression, one would like to deduct the information from the original embedding that is related to the specific attribute to be suppressed. So, the original x and the suppressed x' can not be similar.

2. the formulation in Eqn. (9) allows both multiple attributes for suppression and multiple attributes for reservation. However, across all the experiments, the demonstration is only one attribute for suppression. This can hardly justify the scalability of the method.

3. for the joint loss, there is loss weights designed. However, there is no empirical value or suggested value for the replication.

4. In the ablation study, those loss terms are incrementally increasing. However, it is not clear enough to highlight each of the loss component. For example, in table 2, would all loss terms but without $L_{r^{\*}}$ a valid ablation baseline? This is to highlight how important the $L_{r^{\*}}$ w.r.t. the joint loss.

5. Across all the experiments, the proposed method is mostly compared to its ablative baselines. Only Table 5 is compared to CIAGAN on one dataset, Adience.

6. In the experiment, the last second paragraph before "Conclusion" section, the authors mentioned:
"Lastly, comparing to SPAct (Dave et al., 2022) on PA-HMDB1, MaSS achieved competitive suppression ratio of cMAP on both datasets (VISPR: 55.6% vs. 57% and PA- HMDB: 20.7% vs. 16%) for the other 5 non-action attributes."
However, in Table 4, the P-HMDB dataset evaluation, we did not see the comparison to SPAct.



**Strength And Weaknesses:**

Strength:
1. the technical pipeline of the selective suppression is clearly illustrated with adequate figures and formulated equations and losses.

2. the experiments are designed to cover wider range of attribute classifications including face, audio and action recognition.

3. the experiments are towards extensive to check each of the ablative components, and the hyper-parameters for a in-depth comparison for better understanding.

Weakness:
1. the scalability of dealing with many attributes is not sufficiently verified. Through all the evaluation datasets, the attributes considered is less than 10, which cannot directly expand to large scale scenario.

2. the compared methods are mostly focusing on MaSS's ablative variations. There significantly lacks the comparison to other state-of-the-art methods, thus not so clear how better it is against the literature.

**Summary Of The Paper:**

This paper proposes a multi-attribute selective suppression (MaSS) framework for the multi-attribute classification task, where there is some targeted attribute to be suppressed. The authors apply a MLP based Encoder-decoder with skip link architecture to achieve the task, where it is decomposed into mainly three parts, the suppression branch, the data modifier (the architecture), and the preservation branch. Their method is further evaluated across a set of experiments, leveraging the face attribute of gender and age while suppressing ID, the audio sound of MNIST for digit classification while suppressing speaker ID, and the PA-HMDB for action classification while suppressing skin color, gender and other attributes. The overall performance has shown clear improvement over baselines and some SOTA methods.

**Summary Of The Review:**

Overall the paper presents a technically valid framework for interested attribute suppression based on the entropy maximization loss design. However, from the experiment, it does not show the ability to scale up to multiple attributes suppression, neither was it sufficiently compared to state-of-the-art methods to demonstrate its effectiveness. Thus, I would like to see the enrichment of the experiment for the final rating. For now, it is borderline towards reject.

---

> ### Author Response · Authors · 2022-11-19
> **Response to Reviewer HMFn**
>
> We thank the reviewer for recognizing that our experiments "... are designed to cover wider range of attribute classifications," and are designed to be extensive in checking each ablative components. Below please find our point-to-point responses to the reviewer's comments.
>
> **[W1] the scalability of dealing with many attributes.**
>
> In our experimental evaluations, we tested MaSS on three datasets across different domains, with 3, 4, and 6 attributes, respectively, as opposed to only 2 or 3 attributes. We consider the current scale of our experiments and the promising results we obtained a solid indication of scalability of our algorithm. That said, we completely agree with the reviewer's comment that testing MaSS on datasets with a much greater number of attributes (e.g., 10+) would give us a higher confidence in MaSS' scalability. We are currently actively working on this. But given the rebuttal's time constraint, we have to defer the inclusion of such much-larger-scale experiments to future work.
>
>
> **[W2] the compared methods are mostly focusing on MaSS’s ablative variations.**
>
> We thank for reviewer in pointing out our oversight in our original evaluation. In the revision, we added comparative experiments of two SOTA methods, DeepPrivacy [1] and Fawkes [2], on the Adience dataset. As shown by the results in the table below, in suppressing the DataID attribute, these two methods also cause significant degradation to the recognition of the age and gender attributes.
> The new results are added in Table 5 of the manuscript.
>
> **Table: More comparison to other methods on Adience. DataID is selected as the suppression target. Top-1 Accuracy (\%) is reported.**
>
> |  | DataID |  Age | Gender |
> | --- | --- | --- | --- |
> | Original | 90.8 | 89.1 | 97.4 |
> | CIAGAN |  1.1 | 17.8 | 66.9 |
> | DeepPrivacy |  5.9 | 32.2 | 84.0 |
> | Fawkes | 24.7 | 47.6 | 87.8  |
> | Ours |  0.6 | 78.5 | 95.7 |
>
>
> Reference:
>
> [1] Håkon Hukkelås, Rudolf Mester, Frank Lindseth, "DeepPrivacy: A Generative Adversarial Network for Face Anonymization," ISVC 2019.
>
> [2] Shan, Shawn, et al. "Fawkes: Protecting privacy against unauthorized deep learning models." 29th USENIX Security Symposium (USENIX Security 20). 2020.
>
> Below we answer the questions from the Clarity section
>
> **[C1-1] for all the experiments, what exact similarity measure is utilized?**
>
> We use the cosine distance for the to-be-suppressed attributes, NT-Xent for generic feature preservation and KL divergence for the to-be-preserved attributes if the label information is available.
> We described these settings in Appx. B.3. We also updated the experimental setup section to clarify these settings.
>
> **[C1-2] The original x and the suppressed x’ can not be similar.**
>
> As the reviewer mentioned, for suppression, the original data x and the transformed data x' indeed can not be similar with respective to the suppressed attribute(s).
> However, x and x' could still be similar with respective to other attributes.
> This is exactly the goal of our work: suppressing selected attributes while keeping other attributes virtually intact.
>
>
> **[C2] Across all the experiments, the demonstration is only one attribute for suppression.**
>
> We thank the reviewer for pointing out this limitation of our original evaluation. We conducted additional experiments on the Adience dataset, where we targeted two attributes for suppression and the other one attribute for preservation. The result is shown in the table below, which we also added in the appendix (as Table 11). Note that MaSS is configured with $L^{r_*}$ and $L^{r_1}$
>
> **Table: Suppressing multiple attributes on Adience. DataID and Gender are selected as the suppression targets. Top-1 Accuracy (\%) is reported.**
>
> | | DataID ($s_1$) | Gender ($s_2$) | Age ($r_1$) |
> | --- | --- | --- | --- |
> | Original | 90.8 | 97.4 | 89.1 |
> | Ours | 0.9 | 5.4 | 83.8 |
>
> **[C3] For the joint loss, there is loss weights designed**
>
> For most of the experimental settings, we described those details in Appx. B due to the space limit.
> Specifically, the training details of MaSS are in Appx. B.3 and Table 7 illustrates the weights of each loss and each dataset for reproducibility.
> We revised the experimental setup section of the paper to clarify these settings.

---

> > ### Author Response · Authors · 2022-11-19
> > **Response to Reviewer HMFn (Cont.)**
> >
> > **[C4] For example,
> > in table 2, would all loss terms but without $L^{r_{\*}}$ a valid ablation baseline.**
> >
> > As the reviewer suggested, we conducted the experiments without $L^{r_{\*}}$ but all to-be-preserved attributes are given on all three datasets.
> >
> > First, we would like to emphasize that the advantages of using $L^{r_{\*}}$ is that even without any label information in advance, MaSS can still perverse generic features in the transformed data. This is desirable in practice as the data owner might not even know the *complete* set of interesting attributes in advance. But with $L^{r_{\*}}$, the data owner could potentially explore other attributes later.
> > Table 2, 3 and 4 show that MaSS ($L_{rec}+L^{s_1}+L^{r_{\*}}$) achieved good performance without using any label information of the to-be-preserved attributes.
> >
> > The tables below show the additional experiments, which are also included in the latest manuscript (Table 13, 14 and 15.)
> >
> > **Table: [Adience] Ablation studies with $L^{r_*}$ but all to-be-preserved attributes. DataID is selected as the suppression target. Top-1 Accuracy (\%) is reported.**
> >
> > | | With $L^{r_*}$ | DataID | Age | Gender |
> > | --- | --- | --- | --- | --- |
> > | Ours | Yes | 0.6 | 86.9 | 96.7 |
> > | Ours | No | 0.0 | 84.5 | 96.3 |
> >
> > **Table: [AudioMNIST] Ablation studies with $L^{r_*}$ but all to-be-preserved attributes. SpeakerID is selected as the suppression target. Top-1 Accuracy (\%) is reported.**
> >
> > | | With $L^{r_*}$ | SpeakerID | Digit | Accent | Gender |
> > | --- | --- | --- | --- | --- | --- |
> > | Ours | Yes | 1.7 | 99.6 | 95.7 | 98.4 |
> > | Ours | No | 1.6 | 99.7 | 95.0 | 97.7 |
> >
> > **Table: [PA-HMDB] Ablation studies with $L^{r_*}$ but all to-be-preserved attributes. Non-action attributes are selected as the suppression targets. Top-1 Accuracy (\%) is reported for Action and cMAP (\%) is reported for Non-action.**
> >
> > | | With $L^{r_*}$ | VISPR (Non-action) | PA-HMDB (Action) | PA-HMDB (Non-action) |
> > | --- | --- | --- | --- | --- |
> > | Ours | Yes | 38.6 | 58.0 | 63.4 |
> > | Ours | No | 38.3 | 52.2 | 63.6 |
> >
> > Although without $L^{r_{*}}$, it achieved performance on to-be-preserved attributes competitive to both $L^{r_{\*}}$ on Adience, as well as all attributes on AudioMNIST; we see that for PA-HMDB, without $L^{r_\*}$, the accuracy of the action attribute is degraded by 5.8\%, we conjecture that the generic feature retained by $L^{r_\*}$ helps the data utilities of the modified data.
> >
> > **[C5] More comparison on Adience.**
> >
> > As mentioned in **[W2]**, we compared two more methods on Adience and all results show that our method can suppress the selected attribute while retaining other attributes.
> >
> > **[C6] Comparison to SPAct.**
> >
> > We tried our best to compare with their results under the same experimental condition; however, our absolute performance over the original data is significantly better than what they reported in their paper; therefore, we only compare the relative gains with respective to its own baseline. Moreover, under this setting, they do not show the accuracy of the action attribute.
> > We added a new table to compare with SPAct with those details in Appx. D. (Table 16).
> >
> > **Table: Comparison to SPAct with respective to their own baseline. Non-action attributes are selected as the suppression targets. MaSS is configured without using action label. Top-1 Accuracy (\%) is reported for Action and cMAP (\%) is reported for Non-action.**
> >
> > | | VISPR (Non-action) | PA-HMDB (Action) | PA-HMDB (Non-action) |
> > | --- | --- | --- | --- |
> > | Original Data (Ours) | 81.8 | 58.7 | 79.7 |
> > | Ours | 36.3 ($\downarrow$55.6\%) | 52.1 | 63.2 ($\downarrow$20.7\%) |
> > | Original Data (SPAct) | 64.4 | - | 70.1 |
> > | SPAct | 27.4 ($\downarrow$57\%) | - | 58.9 ($\downarrow$16\%) |

---

> > > ### Comment · Reviewer_HMFn · 2022-11-23
> > > **Response to the authors**
> > >
> > > Thanks to the authors for their detailed feedback. While part of my concerns have been addressed, there are still several points that I remain unconvinced.
> > >
> > > 1. the reconstruction of x and x' to be similar is not convincing. By design, one would remove the sensitive attribute out of the current embedding x to obtain x'.
> > >
> > > The authors mentioned "keeping other attributes intact". This is not to be achieved by such reconstruction, but more from the classification losses illustrated in Sec. 3.2.3 preservation branches.
> > >
> > > The above being said, it might be very important to ablate on L_rec w/ and w/o such item and observe the evaluation behavior.
> > >
> > > 2. the ablation L_{r_*} tentatively suggests that it is not as important as the other modules, and to some extent, it is negligible in terms of the performance difference.
> > >
> > > 3. the additional comparison to SPAct is not decisive, i.e., for attribute to be suppressed, SPAct is better (27.4 to 36.3), while for attribute to be preserved, SPAct is worse (58.9 to 63.2).
> > >
> > > Based on the above concerns, I probably cannot raise my rating at the current moment.

---

> > > > ### Author Response · Authors · 2022-11-30
> > > > **Authors' Response for Reviewer HMFn**
> > > >
> > > > Please see our response below.
> > > >
> > > > 1. The reconstruction loss is merely a generic L2 regularizer within the suppresion branch. The main mechanism that does the actual heavy-lifting in shaping the x->x' transformation is the adversarial interaction between the suppression branch and the preservation branch, where the former aims to make x and x' differ **in terms of** their classification results on the suppressed attributes, and the latter is tasked to keep x and x' similar **in terms of** their classification results on the rest of the attributes, especially those we explicitly want to preserve.
> > > >
> > > > 2. We respectively disagree with the reviewer's comment that $L^{r_*}$ is not as important as the other modules. Our MaSS framework, as well as our experimental settings, were designed to account for various scenarios with different realistic information-availability constraints. There are certainly plausible situations where the data owner knows in advance exactly what specific attributes ($r$'s) to preserve, in which cases they can use the corresponding attribute-specific losses in their computation; But we can also imagine situations where a data owner might not know in advance what specific set of attributes to preserve -- for example they simply only want to suppress the sensitive attributes (e.g., identity) and otherwise leave the data as intact as possible for any potential future downstream tasks, which the data owner could not possibily have predicted in advance. In this latter case, $L^{r_*}$ is all the data owner has, but still leads to very promising results. For example, as shown in Table 2, the first row for MaSS indicates that, when suppressing DataID, if we know nothing in advance about the Age and Gender attributes and only use the reconstruction loss, Age and Gender are both significantly damaged. The next row (second row for MaSS) then illustrates, by using $L^{r_*}$, we dramatically reduce the degradation to the Age and Gender attributes caused by the data transformation, which shows the importance of $L^{r_*}$. The next two rows then further demonstrate that if more information is available, our solution of course is able to take advantage of them and further improve the suppression+preservation performance. Similar trends are also observed in Table 3 and 4.
> > > >
> > > >
> > > > 3. In the table (also Table 16 in Appendix), both columns (VISPR non-action and PA-HMDB non-action) are the **suppressed** attributes while the action column is the **preserved** attribute. Please note that the SPAct paper does not provide their action recognition accuracy.
> > > > As can be seen, for the non-action attributes on both VISPR and PA-HMDB, MaSS achieves competitive suppression results against SPAct in terms of recognition reduction ratio (MaSS' 55.6\% vs SPAct's 57\% for VISPR; MaSS' 20.7\% vs SPAct's 16\% for PA-HMDB). (Note that: **Original (Ours)** is the baseline we produced and **Original (SPAct)** is the baseline reported in their paper.)
> > > > Furthermore, as discussed in our paper, SPAct is designed to suppress non-action attributes only but MaSS is much more flexible and fully configurable.

---

### Author Response · Authors · 2022-11-19
**Summary of Authors' Response and Paper Revision**

We thank all the reviewers for their very constructive feedback on our paper! We are encouraged by the fact that the reviewers find that
*i*) our paper is well-written, easy to follow, and with clearly illustrated technical details [HMFn, 8b8w, uwsC];
*ii*) the research topic is helpful [8b8w], with valid and important motivation [SL3b, uswC];
*iii*) the proposed method is straightforward, flexible and effective [SL3b];
*iv*) our experimental evaluation covers a wide range of attribute classifications across multiple domains [HMFn, 8b8w]; and
*v*) the ablative experiments are extensive [HMFn] and the performance is effective [uwsC].

We have made every effort to best address all the concerns that the reviewers posed with additional experiments, comparisons, and clarifications. Each and every corresponding edits and additions are discussed in detail in this response document, as well as highlighted in red in the latest updated manuscript. Below, please find our revision summary:

1. Additional comparison to DeepPrivacy [1] and Fawkes [2] on Adience. [HMFn, uwsC]
2. Additional results on suppressing multiple attributes [HMFn]
3. Additional results on studying the losses [HMFn, SL3b]
4. Clarifying how to pretrain the models used in MaSS, as well as how to train MaSS itself, including hyperparameters, loss weights, etc. [8b8w, SL3b, uwsC]


Reference:

[1] Håkon Hukkelås, Rudolf Mester, Frank Lindseth, "DeepPrivacy: A Generative Adversarial Network for Face Anonymization," ISVC 2019.

[2] Shan, Shawn, et al. "Fawkes: Protecting privacy against unauthorized deep learning models," 29th USENIX Security Symposium, 2020.

---

### Decision · Program_Chairs · 2023-01-20

**Decision:**

Reject

**Justification For Why Not Higher Score:**

This paper presents yet another privacy approach without clear privacy guarantees, rendering the approach completely useless and potentially dangerous because it is misleading.

**Justification For Why Not Lower Score:**

N/A

**Metareview: Summary, Strengths And Weaknesses:**

The paper studies an approach that enables removal of specific attributes from data while preserving the remaining features of the data for downstream machine learning applications. The paper evaluates the efficacy of the proposed approach in experiments that suggest removed attributes are difficult to infer while other machine-learning predictions can still be performed quite well.

The reviewers are generally lukewarm about the paper, expressing around scalability of the approach, comparisons with alternative methods, and clarity of the presentation. While some of these concerns were addressed in the discussion phase, all reviewers remain lukewarm about the paper.

For that reason, the AC decided to read the paper themselves. They noticed a severe weakness in the paper that was not mentioned by the reviewers. While the paper focuses on preserving privacy of some aspects of the data, the paper does not formalize the privacy definition or threat model it adopts. As a result, it is unclear what guarantees the paper actually provides. For example, is it truly impossible to infer DataID in the Adience dataset? Or is it just not possible to infer DataID with the model that was used, but would another model family infer it just fine? Without rigorous privacy guarantees, it is impossible to answer this questions as a result of which the true value of the proposed approach is difficult to gauge. The proposed approach would have much more real-world relevance if it provided an attribute-level version of privacy guarantees on membership inference (differential privacy) or data reconstruction (Fisher information loss). Earlier methods without such guarantees were rapidly broken rendering them useless (for example, InstaHide was broken by Nicolas Carlini) and the approach proposed in this paper may suffer a similar faith without robust privacy guarantees.